# Exploring Applications of Artificial Intelligence in Critical Care Nursing: A Systematic Review

**DOI:** 10.3390/nursrep15020055

**Published:** 2025-02-04

**Authors:** Elena Porcellato, Corrado Lanera, Honoria Ocagli, Matteo Danielis

**Affiliations:** 1Laboratory of Studies and Evidence Based Nursing, Department of Cardiac, Thoracic, Vascular Sciences and Public Health, University of Padova, Via Loredan 18, 35131 Padova, Italy; porcellatoelena@gmail.com; 2Unit of Biostatistics, Epidemiology and Public Health, Department of Cardiac, Thoracic, Vascular Sciences and Public Health, University of Padova, Via Loredan, 18, 35131 Padova, Italy; corrado.lanera@ubep.unipd.it (C.L.); honoria.ocagli@ubep.unipd.it (H.O.)

**Keywords:** artificial intelligence, critical care nursing, outcomes, systematic review

## Abstract

**Background:** Artificial intelligence (AI) has been increasingly employed in healthcare across diverse domains, including medical imaging, personalized diagnostics, therapeutic interventions, and predictive analytics using electronic health records. Its integration is particularly impactful in critical care, where AI has demonstrated the potential to enhance patient outcomes. This systematic review critically evaluates the current applications of AI within the domain of critical care nursing. **Methods:** This systematic review is registered with PROSPERO (CRD42024545955) and was conducted in accordance with PRISMA guidelines. Comprehensive searches were performed across MEDLINE/PubMed, SCOPUS, CINAHL, and Web of Science. **Results:** The initial review identified 1364 articles, of which 24 studies met the inclusion criteria. These studies employed diverse AI techniques, including classical models (e.g., logistic regression), machine learning approaches (e.g., support vector machines, random forests), deep learning architectures (e.g., neural networks), and generative AI tools (e.g., ChatGPT). The analyzed health outcomes encompassed postoperative complications, ICU admissions and discharges, triage assessments, pressure injuries, sepsis, delirium, and predictions of adverse events or critical vital signs. Most studies relied on structured data from electronic medical records, such as vital signs and laboratory results, supplemented by unstructured data, including nursing notes and patient histories; two studies also integrated audio data. **Conclusion:** AI demonstrates significant potential in nursing, facilitating the use of clinical practice data for research and decision-making. The choice of AI techniques varies based on the specific objectives and requirements of the model. However, the heterogeneity of the studies included in this review limits the ability to draw definitive conclusions about the effectiveness of AI applications in critical care nursing. Future research should focus on more robust, interventional studies to assess the impact of AI on nursing-sensitive outcomes. Additionally, exploring a broader range of health outcomes and AI applications in critical care will be crucial for advancing AI integration in nursing practices.

## 1. Introduction

Artificial intelligence (AI) has been defined by the high-level expert group on AI (AI HLEG) as “a system designed by humans that act in the physical or digital world by deciding the best action(s) to take, according to predefined parameters, to achieve the given goal” [1]. AI has been involved in different settings such as economics, industries, and healthcare. In healthcare, AI has been applied to medical imaging, personalized diagnostics, and therapeutics, as well as to the development of predictive models through the analysis of data from electronic medical records [2]. The development of AI and its application in medicine can play an important role in the transformation of healthcare settings [3].

AI has been used in intensive care and emergency settings for diagnostic and prognostic predictive modeling in the context of coronavirus disease 2019 (COVID-19) [4]. Mortality predictions have also been analyzed in cardiac intensive care unit patients, with AI applied to electrocardiograms (ECGs). The AI algorithm for ECG interpretation (AI-ECG) demonstrated the ability to detect left ventricular systolic dysfunction with an accuracy of 76%, a sensitivity of 73%, and a specificity of 78%. Additionally, the AI-ECG showed the ability to predict 1-year mortality in patients with an ECG interpreted by a cardiologist, achieving an AUC of 0.876 [5].

AI has also been implemented to analyze topics specific to the intensive care unit (ICU), such as mechanical ventilation and prediction of extubation. These outcomes have been specifically examined using AI models to improve clinical decision-making in ICU settings [6,7]. AI was able to identify key factors significantly associated with extubation, including the duration of ventilation, rapid shallow breathing index, and Glasgow Coma Scale [7]. AI has also been used in trauma triage, with AI algorithms compared to the best-performing triage tools for predicting mortality. A meta-analysis indicated that AI models significantly outperformed conventional triage tools in predicting mortality, achieving an AUC-ROC of 0.09 [8].

When examining nursing-sensitive outcomes, AI has shown potential to improve the identification, prevention, and management of patient conditions, thus enhancing patient care and nursing practice [9]. For instance, the objective of Alderden’s study [9] was to develop a model that could predict the development of pressure injury. The model uses information directly from electronic health records; in this way, nurses do not need to put information into an external tool such as the Braden scale.

A 2024 study by Hassan and El-Ashry investigated how critical care nurses make sense of the challenges and opportunities associated with leading with AI in their practice [10]. AI techniques in critical care offer significant advantages, including the real-time data analysis for the early detection of patient deterioration, automation of routine tasks, and augmentation of clinical expertise. These capabilities allow nurses to focus on patient-centered care, improve decision-making, and enhance collaboration across healthcare teams. However, challenges remain [10]. The overabundance of AI-generated alerts can lead to information overload, while the lack of transparency in some algorithms undermines trust. Initial implementation phases often increase workload due to training and technical adjustments. Ethical concerns, such as biases in AI systems and risks to patient autonomy, further complicate integration. Additionally, overreliance on AI may erode critical thinking skills if not carefully managed [10].

To date, AI has garnered significant attention in critical care, with numerous studies exploring its potential and applications. However, a comprehensive synthesis is lacking that clearly identifies which AI techniques have been effectively implemented in critical care settings, the metrics used to evaluate them, the types of data processed by AI systems, and the nursing outcomes considered. Such an overview would not only provide a solid foundation for designing future research but also guide the development of more targeted and effective applications, ultimately enhancing clinical practice and nursing care in critical care environments.

The aim of this systematic review is to analyze the application of AI in critical care nursing, focusing on outcomes specific to nursing care. Specifically, the review explores the AI model techniques applied in critical care settings such as the ICU, emergency department (ED), operating room (OR), recovery room, and prehospital emergency care.

## 2. Materials and Methods

### 2.1. Study Design

This is a systematic review registered with the PROSPERO International Prospective Register of Systematic Reviews (CRD42024545955). Searches were conducted on MEDLINE (Medical Literature Analysis and Retrieval System Online) via PubMed, SCOPUS, CINAHL (Cumulative Index in Nursing and Allied Health Literature), and Web of Science, with papers restricted to the English and Italian languages. The publication period covers studies from the inception of each database up to 30 June 2024. No time limitations were imposed as the phenomenon under study is relatively recent, and the aim was to include all possible relevant studies.

The main outcomes were as follows: (a) the techniques used to train AI models; (b) the critical care settings (ICU, ED, OR, recovery room, prehospital emergency) in which AI was applied; (c) the clinical aspects addressed; and (d) the types of data utilized in the models.

The review adhered to the PRISMA (Preferred Reporting Items for Systematic Reviews and Meta-Analyses) guidelines (Appendix A) [11].

### 2.2. Inclusion and Exclusion Criteria

Studies were included in the review if they met the following inclusion criteria: the study applied an AI model to critically ill adult patients (aged 18 years or older), was conducted in a critical care setting, and addressed topics or contexts relevant to nursing practice. Eligible study designs included observational, experimental, qualitative, and case studies. Secondary studies, such as reviews or meta-analyses, were excluded.

The exclusion criteria were as follows: studies that did not employ any specific AI model training technique or did not report the technique used, and studies focusing on AI applications in contexts outside of critical care settings.

### 2.3. Study Selection

The search strategy for each database was structured around three key macro themes, connected by the Boolean operator “AND”. In this strategy, “OR” is used to connect alternative keywords within each theme. The first theme covered critical and acute care settings (e.g., “emergency department”). The second focused on AI technologies such as machine learning, deep learning, and natural language processing. The third emphasized nursing in these settings (e.g., “critical care nursing”). By combining these themes, the strategy captured interdisciplinary studies on integrating AI into nursing practice in critical care environments. The complete search strategy for each database is provided in Appendix A.

The Covidence platform (Covidence systematic review software, Veritas Health Innovation, Melbourne, Australia, 2024. Available at www.covidence.org, accessed on 1 July 2024) was used to manage and organize the data. This facilitated the accurate screening of articles and the elimination of duplicates in multiple databases [12].

The selection of studies for inclusion, first based on title and abstract and subsequently on full text, was performed independently by two reviewers (E.P. and M.D.). Conflicts were resolved through discussion with the involvement of a third reviewer (H.O.).

### 2.4. Data Extraction

Data were extracted using an Excel form, initially completed by one reviewer and then verified by a second reviewer. The form was modified as needed based on the reviewers’ feedback. The following data were extracted and recorded from the included studies: general study information (title, author(s), year, DOI, location, design); sample details (general sample, training, validation, and test samples, if available); study objectives and main findings; and the phenomenon of interest (technology used, AI application including name, validation, and performance, as well as the application theme).

### 2.5. Studies’ Quality Assessment

The methodological quality of quantitative, qualitative, and mixed-method studies was independently assessed by two authors using the Mixed Methods Appraisal Tool (MMAT) developed by Hong et al. [13]. The methodological quality score was not a criterion for excluding studies. The quality assessment is detailed in Appendix A.

### 2.6. Knowledge Synthesis

Given the descriptive nature of the research question and the methodological heterogeneity of the included studies, data were analyzed using a narrative approach and categorized following a logical structure to address the research questions. The extracted data were systematically tabulated to provide a comprehensive overview. The synthesis included the general characteristics of the included studies, total sample sizes, training models and techniques, types of artificial intelligence models, performance measures, and data types. This structured approach facilitated the identification of patterns and key findings across the selected studies.

## 3. Results

### 3.1. General Characteristics

In the initial literature review, 1364 articles were identified. After removing 507 duplicates, 857 articles were screened. Ultimately, 24 articles were included in the systematic review. A PRISMA flow diagram of the included studies is provided (Figure 1). The studies were conducted in different countries, including the USA (8), Belgium (1), Netherlands (1), Italy (1), Israel (2), South Korea (3), Japan (1), Taiwan (1), China (2), Spain (1), and Brazil (2); the country of one study was not specified. Of the included studies, only one had a qualitative design, while the rest were quantitative, primarily observational or retrospective in nature.

The critical care settings described in the studies include the ICU, with 13 studies, and the ED, with 9 studies. Perioperative care was addressed in one study, and prehospital emergency care in another. Various health outcomes were analyzed across the studies, such as postoperative complications, ICU admission or discharge, triage level/triage code assessment, pressure injuries, sepsis, prediction of adverse events, delirium, and critical vital signs. The general characteristics of the studies are summarized in Table 1 and Table 2.

The studies applied AI to various sample sizes, which are presented in the table below. One article was excluded due to its qualitative design [14]. The range of min–max is 2 [14]–560,486 [16].
nursrep-15-00055-t002_Table 2Table 2Total sample sizes of studies.Total Sample Size Number of Studies <1002100–100041000–10,000610,000–100,0006>100,0005


### 3.2. Training Models’ Techniques

All studies utilized various training techniques, categorized into deep learning techniques, machine learning techniques, Large Language Models, and classical techniques. Deep learning techniques encompass all types of neural networks, while classical models include methods like logistic regression. Machine learning techniques cover approaches other than neural networks or classical models, such as random forest, support vector machines, and XGBoost. Large Language Models, including ChatGPT, were also considered as a training model technique. Specifically, ChatGPT, a chatbot developed through supervised reinforcement learning strategies, was applied to triage in the study by Zaboli et al. [19]. Classical models are distinguishable as they are parametric, using techniques as such as logistic regression. In contrast, machine learning models are either non-parametric or utilize shallow neural networks with multiple nodes—for example, decision tree, random forest, or XGBoost—or one-hidden-layer fully connected neural network. Deep learning models, in contrast, are based on neural networks that include at least two hidden layers [1].

Most studies analyzed and applied different training models to identify the one that performed best for the specific outcome. The performance of these techniques was compared using various metrics, including accuracy, sensitivity, precision, negative predictive value, positive predictive value, and Receiver Operating Characteristic Area Under the Curve (AUC-ROC). The training phase is the initial stage where the AI model is applied to a dataset, learning from the data by identifying patterns and relationships while adjusting its parameters to minimize prediction errors. After training, the model enters the validation phase, where its performance is assessed, allowing for fine-tuning and improvements. This iterative process is crucial to enhance the model’s robustness. Finally, in the test phase, the fully developed model is applied to a new, unseen dataset, with the results being considered definitive, providing an unbiased evaluation of the model’s practical performance [38]. Some studies emphasized the distinctions between the training, validation, and testing phases, some focusing on the performance of the best model in all three stages [21,33,36]. On the contrary, others examined only the test [19,37] or the training phase [16,26,27,35]. Performance measures can vary depending on the phase; for instance, models are often evaluated with fewer metrics during training compared to testing [33], or they may use different measures together [18]. The population sample was typically divided into training, validation, and test subsets. Some studies only described the training sample [16,35]; while others allocated samples into training and testing phases using various ratios, such as 80:20 or 70:30 [18,23]. The main training model techniques are summarized in Table 3, along with the performance measures applied to each of them (Appendix A).

### 3.3. Type of Data Approach

AI requires data to be analyzed to achieve the desired results. Data can be classified into structured and unstructured types. Structured data include continuous, categorical, and binary data, while unstructured data can include images, text, and audio. AI is capable of analyzing and extracting information from all these data types. Most studies analyzed structured data or text (considered unstructured data), which are commonly found in electronic medical records (EMRs). Examples of structured data include vital signs, demographic data, laboratory results, and comorbidities. Text data, typically nursing notes, medical histories, and medication records, is a common form of unstructured data analyzed by AI. Audio data have been used in AI applications in only two studies; in one study, a voice AI was employed to record information during triage in the ED [37]. In the second study, a digital assistant was able to analyze real-time discussions during rounding teams [29]. No other unstructured data have been applied. AI is utilized to analyze different types of data, with the output varying based on the primary outcome of the AI application. For example, if the main outcome was triage code assessment, the output would be categorical data. The input and output data types used in the studies, along with the corresponding AI applications, are summarized in Table 4 and Appendix A.

## 4. Discussion

This systematic review offers a comprehensive overview of the application of AI in critical care nursing settings, examining the nature of the data used, the methods employed, and the associated clinical and nursing outcomes, while highlighting AI’s impact on patient care and its evolving role in clinical decision-making. Machine learning emerged as the most widely used method, with the most applied techniques being gradient-boosted trees, decision trees, random forests, and support vector machines [14,15,16,17,18,20,21,22,24,26,31,36]. Machine learning techniques are highly adaptable and can be applied to a wide range of sample sizes. The growing use of machine learning in healthcare has been particularly beneficial when applied to large patient datasets, enhancing the study outcomes. Notably, studies that use machine learning can handle sample sizes exceeding 100,000 individuals, enabling the generation of more robust and generalizable findings [16,17,21,22,24]. However, machine learning techniques can also be applied effectively to smaller populations, less than 100,000, still providing valuable insights and meaningful results [15,18,20,26,31,36]. On the contrary, traditional methods such as logistic regression typically handle a sample size of up to 10,000 individuals [27,33], which limits their ability to capture the complexity and variability of diverse patient populations. Other studies applied traditional technologies up to a population of 1000 individuals [29,32]. Only one study applied classic techniques to a larger population, less than 100,000 [23]. Deep learning techniques have been applied to populations that typically range from 10,000 to 100,000 people [25,30,35]. Only 2 studies involved populations of fewer than 10,000 people [28,37], and 1 study had a population of fewer than 100 people [34]. However, it should be noted that in the totality of the studies, fewer models ultimately applied deep learning as their definitive technique compared to other techniques. This disparity in sample size capabilities underscores the advantage of machine learning in harnessing large datasets to uncover insights that can improve healthcare outcomes and refine treatment strategies.

Among the articles reviewed, the predominant focus is on the development of predictive models for various health outcomes. Researchers are increasingly leveraging machine learning techniques to create models that can predict disease progression and patient outcomes in conditions such as acute coronary syndrome, sepsis, and high-pressure injuries. These models aim to enhance clinical decision-making and improve patient management in critical care settings [18,20,21,23,32,33,35]. This emphasis on predictive modeling not only deepens our understanding of health dynamics but also facilitates personalized medicine by enabling healthcare providers to tailor interventions based on individual risk profiles. Furthermore, these models act as decision support tools for healthcare professionals, particularly in critical areas such as triage, helping to prioritize care and improve patient outcomes [16,19,22,36,37], as well as addressing more organizational aspects like patient discharge or the admission processes of patients [15,17]. By integrating predictive analytics into clinical workflows, healthcare professionals can make more informed decisions, ultimately improving patient care and operational efficiency.

Another aspect to consider when choosing AI technology is the processing time for data analysis. This effect was analyzed as a performance value in the study by Nazzal et al. [17]. It is an aspect to pay attention to, as the time required for data analysis can vary among different AI technologies; thus, some technologies can provide answers more quickly than others. This aspect can be a key factor in the choice between two different technologies in clinical settings.

In the realm of AI technologies, the data used predominantly consist of tabular data, which can be classified as continuous, binary, or categorical. This includes vital parameters, aspects of the patient’s medical history, laboratory test results, and assessment scales such as the Braden scale. In addition, information from nursing notes has been incorporated to provide a more comprehensive view of patient health. However, there is a noticeable lack of other data types in the reviewed studies. For example, audio data have been scarcely utilized, and no studies applied image data in their models [29,37]. This reliance on structured tabular data highlights both the strengths and limitations of current artificial intelligence applications in healthcare. It suggests a potential area for future exploration, where integrating more diverse data types, such as audio, image, or unstructured text, could enhance model accuracy, applicability, and overall effectiveness in clinical settings. The combination of structured and unstructured data presents a unique challenge to AI model development. Structured data, with their standardized format, allows for straightforward integration and analysis, whereas unstructured data, such as free-text nursing notes or audio recordings, offer context-rich insights that are harder to process and standardize using classical techniques. This heterogeneity can impact the consistency and reliability of model performance. A wealth of information is stored in unstructured data, but their lack of a predefined structure makes it difficult for AI systems to process and analyze efficiently. These data often require significant preprocessing and cleaning to extract meaningful insights, and their variability can lead to inconsistencies that affect the accuracy of AI models. Furthermore, the computational resources needed for processing unstructured data can be substantial, limiting scalability and making their use in AI applications more challenging. Standardizing preprocessing protocols for unstructured data, including text tokenization, audio transcription, or image annotation, could mitigate these challenges to allow researchers and clinicians to apply a wider range of models to their data. Additionally, developing interoperable data formats and guidelines for harmonizing diverse data types would enhance clinical applicability and integration into workflows.

Alongside the beneficial implications explored in this review of AI models, concerns exist regarding their limitations, particularly the reliance on data quality and the potential for bias in algorithms. Critical care nurses interviewed in an interpretive phenomenological analysis by Hassan and El-Ashry identified several ethical dilemmas associated with the use of AI in critical care [10]. Ethical issues with AI in critical care include the possibility of algorithmic bias, which could result in unjust treatment outcomes if the data are not diverse. Another problem is accountability for AI-driven judgments, which require precise protocols to establish who is responsible. Furthermore, an over-reliance on AI runs the risk of undermining clinical expertise, underscoring the necessity of continual training. Lastly, in order to guarantee successful integration into patient care, user-friendly interfaces and open communication are necessary due to the complexity of AI output and the possibility of information overload.

Lastly, while each method offers unique capabilities, a deeper analysis of their specific advantages and limitations is necessary to fully understand their effectiveness in various nursing contexts. This would enable more-informed decisions about which AI techniques are best suited for particular clinical applications, considering factors such as data requirements, interpretability, and resource constraints. For instance, deep learning excels with high-dimensional data but requires significant computational resources and large datasets [35]. In contrast, machine learning methods such as decision trees are more efficient with smaller datasets and easier to interpret, making them suitable for resource-limited settings [26]. Certain models are better suited for specific tasks; for instance, traditional and machine learning methods often perform well in triage assessment and pressure injury prediction [16,23,32,33,36]. NLP techniques applied to nursing notes can uncover subtle clinical patterns [20,22].

### 4.1. Implications for Nursing Practice

The findings of this review offer valuable insights for nursing practice by providing a comprehensive understanding of AI techniques applied in critical care nursing settings. By gaining knowledge of these methodologies, nurses can more effectively evaluate and implement AI-driven models to improve various health outcomes. This understanding allows nurses to use AI to enhance patient monitoring, decision-making processes, and care plans tailored to individual patient needs. Additionally, awareness of AI applications empowers nurses to engage in interdisciplinary collaborations, ensuring that technology is used to its full potential to improve patient care and outcomes. Ultimately, this review serves as a crucial resource for nurses looking to integrate advanced technologies into their practice, fostering a more informed and innovative approach to patient care in critical care settings.

The practical implications for nurses may be that AI tools can significantly enhance nursing practice across critical care areas when integrated appropriately. AI models can predict pressure injury risks in ICU patients by analyzing electronic health records like mobility status and skin assessments, allowing timely preventive interventions. AI techniques applied to nursing documentation and electronic health records can identify high extubation risks, enabling enhanced monitoring protocols. In EDs, AI can recommend accurate triage levels by analyzing patient data, improving prioritization and flow. Voice AI can assist with documentation during triage and rounding, reducing manual entry burdens while prompting evidence-based practices. The implication is that AI can offload routine tasks, enabling nurses to focus more on critical thinking, education, and hands-on care. However, successful implementation requires training nurses on using AI tools effectively while addressing potential algorithm biases. With the right approach, AI can serve as a powerful enabler for high-quality, efficient patient care delivery by nurses.

### 4.2. Strengths and Limitations

The strengths of this review lie in its unique focus on comparing and evaluating AI techniques applied in critical care settings, offering valuable insights for healthcare professionals who may not be well-versed in AI but are eager to understand its applications in these high-stakes environments. By providing a clear overview of the most commonly used methods, the review serves as a useful resource for practitioners who want to enhance their knowledge and apply AI effectively in their work.

The main limitation of this review is the heterogeneity of the included studies, which allows only a narrative synthesis of the results. This diversity in study designs and methodologies limits the ability to draw definitive conclusions about the effectiveness of AI applications in critical care nursing. More robust studies, particularly interventional trials, are needed to test the impact of AI on nursing-sensitive outcomes in critical care settings. These future studies should aim to provide more conclusive evidence on the benefits and challenges of integrating AI into nursing practice. Lastly, although the wide range of AI techniques highlights the adaptability and potential of AI in this field, the review falls short in thoroughly examining the comparative strengths and weaknesses of these approaches. Moreover, the review does not explore why specific models were chosen or compare their performance metrics. Addressing this gap could provide critical insights for practitioners seeking to adopt AI in nursing.

## 5. Conclusions

In conclusion, AI has the potential to be a valuable tool in analyzing the data generated by nurses in their daily practice. By leveraging these data, AI can help develop models that enhance clinical practice and advance research in specific areas. The choice of AI technologies can often be aligned with the ultimate goal or the type of data being analyzed. This review underscores the wide array of AI applications that can contribute significantly to the advancement of nursing practices and research initiatives, offering diverse opportunities to improve patient care and outcomes.

To fully leverage AI’s capabilities, nurses should engage in continuous training and interdisciplinary collaboration to ensure effective implementation. For clinical practice, adopting AI models tailored to specific health outcomes can improve care and efficiency, while also addressing potential biases in AI algorithms. Future research should focus on larger, interventional studies to evaluate the impact of AI on nursing-sensitive outcomes, and explore a broader range of health outcomes to further integrate AI into critical care.

## Figures and Tables

**Figure 1 nursrep-15-00055-f001:**
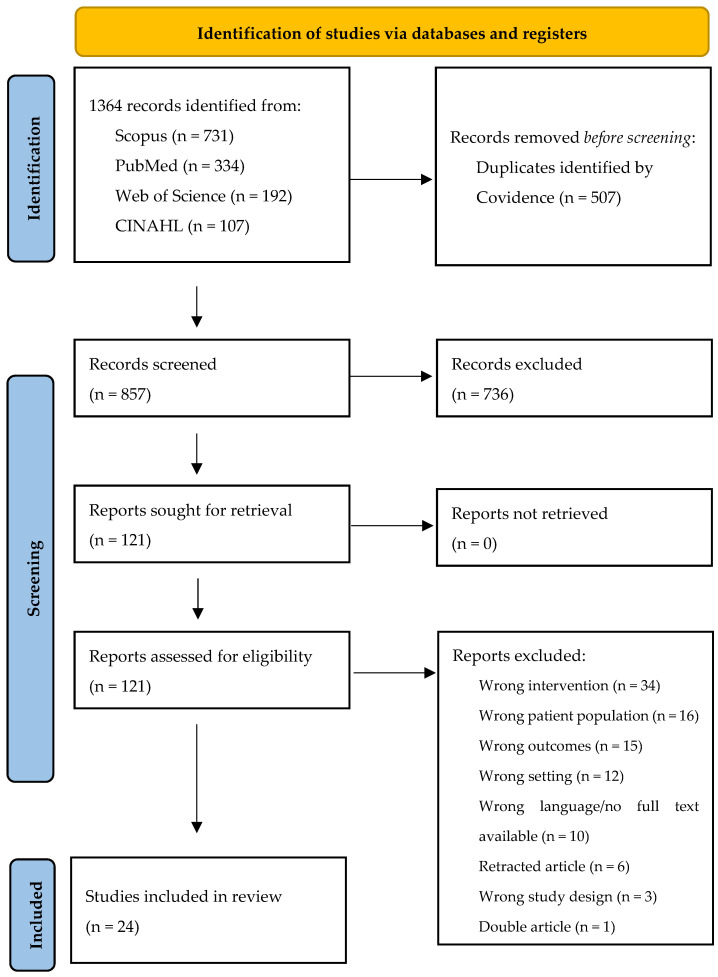
PRISMA diagram of study selection and inclusion.

**Table 1 nursrep-15-00055-t001:** General characteristics of included studies.

Author(s) and Year	Country	Study Design	Health Outcomes	Critical Care Setting
Abraham et al., 2023 [14]	USA	Qualitative study	Postoperative complications (delirium, AKI, DVT, pneumonia, pulmonary embolism)	ICU, OR (perioperative setting)
Meyfroidt et al., 2011 [15]	Belgium	Observational study	Intensive care unit discharge: second day discharge prediction and day of discharge.	ICU
Mutegeki et al., 2023 [16]	USA	Observational study	Triage level: emergency severity index	ED
Nazzal et al., 2020 [17]	USA	Observational study	Intensive care unit admission	ICU
De Koning et al., 2023 [18]	Holland	Retrospective cohort study	Prediction of acute coronary syndrome	Prehospital care, ED
Zaboli et al., 2024 [19]	Italy	Observational study	Triage code assessment	ED
Gu et al., 2024 [20]	USA	Observational study	High-pressure injuries	ICU
Horgn et al., 2017 [21]	Israel	Retrospective, observational cohort study	Sepsis	ED
Ivanov et al., 2021 [22]	USA	Retrospective study	Triage level: emergency severity index	ED
Cramer et al., 2019 [23]	Israel	Observational study	Prevention of pressure ulcers	ICU
Greenbaum et al., 2019 [24]	not reported	Retrospective cohort before-and-after study design and qualitative study	Prediction of presenting problems in emergency department	ED
Yu et al., 2020 [25]	Korea	Retrospective study	Prediction of adverse clinical outcome (mortality in the emergency department or intensive care unit admission)	ED
Tsumoto et al., 2006 [26]	Japan	Observational study	Prediction of adverse events	ED
Lee et al., 2018 [27]	Korea	Observational study	Prevention of unplanned extubation	ICU
Wang et al., 2023 [28]	Taiwan	Retrospective cohort study	Prediction of delirium	ICU
King et al., 2023 [29]	USA	Retrospective cohort study	Integration of evidence-based practices in rounding teams	ICU
Chang et al., 2019 [30]	USA	Observational study	Prediction of critical vital signs	ICU
Bhattacharyya et al., 2022 [31]	USA	Observational study	Delirium prediction	ICU
Xu et al., 2022 [32]	China	Retrospective cohort study	Prediction of pressure injury	ICU
Ladios-Martin et al., 2020 [33]	Spain	Retrospective study and sequential prospective study	Prediction of pressure injury	ICU
Toledo et al., 2024 [34]	Brazil	Observational study	Bed bath time	ICU
Cui & Jin, 2023 [35]	China	Observational study	Prediction of pressure injury	ICU
Brandao-de-Resende et al., 2023 [36]	UK, Brazil	Retrospective study	Triage level	ED
Cho et al., 2022 [37]	Korea	Prospective interventional study	Triage	ED

Abbreviations: AKI = Acute Kidney Injury; DVT = deep vein thrombosis; ICU = intensive care unit; OR = operating room; ED emergency department.

**Table 3 nursrep-15-00055-t003:** The training models’ techniques, model of AI, and performance measures. AUC-ROC = Receiver Operating Characteristics Area Under the Curve; AUPRC = area under precision recall curve.

Author and Year	Training Model Technique(s)	Techniques of AI	Performance’s Measure(s)
Abraham et al., 2023 [14]	Gradient-boosted tree model	Machine learning	Accuracy Predicted risk
Meyfroidt et al., 2011 [15]	Gaussian Processes (GP)	Machine learning	AUC-ROC Brier score Hosmer–Lemeshow u-statistic Loss penalty function Root mean square relative error
Mutegeki et al., 2023 [16]	Decision tree classifier Random forest classifier XGBoost Hist gradient boosting	Machine learning Machine learning Machine learning Machine learning	Precision Accuracy F-1 score AUC
Nazzal et al., 2020 [17]	Naive bayes Generalized linear model Logistic regression Fast large margin Deep learning Decision tree classifier Random forest classifier Gradient-boosted trees Support vector machine	Machine learning Machine learning Classic model Deep learning Machine learning Machine learning Machine learning Machine learning Machine learning	Accuracy Precision Sensitivity Specificity Processing time (min)
De Koning et al., 2023 [18]	Support vector machine Random forest classifier K-nearest neighbor Logistic regression	Machine learning Machine learning Machine learning Machine learning	Sensitivity Specificity Negative predictive value Positive predictive value Precision F-beta score
Zaboli et al., 2024 [19]	ChatGPT	Large Language Model	Sensitivity Specificity Negative predictive value Positive predictive value AUC-ROC Unweighted Cohen’s kappa
Gu et al., 2024 [20]	Logistic regression XGBoost	Classic model Machine learning	Accuracy AUC Precision Sensitivity F-1 score
Horgn et al., 2017 [21]	Support vector machine Logistic regression Naive bayes Random forest	Machine learning Classic model Machine learning Machine learning	AUC-ROC Sensitivity Specificity Positive predictive value
Ivanov et al., 2021 [22]	XGBoost	Machine learning	Accuracy AUC-ROC F-1 score Precision Sensitivity Under-triage Over-triage
Cramer et al., 2019 [23]	Logistic regression Elastic net Support vector machine with a linear kernel Random forest Gradient boosting machine Feed-forward neural network with a single hidden layer	Classic model Machine learning Machine learning Machine learning Machine learning Deep learning	Precision Sensitivity
Greenbaum et al., 2019 [24]	Support vector machine	Machine learning	Structured data capture Completeness Precision Overall quality Mean keystrokes required
Yu et al., 2020 [25]	Logistic regression Deep learning with R package “keras” Random forest	Classic model Deep learning Machine learning	AUC-ROC
Tsumoto et al., 2006 [26]	Decision tree classifier	Machine learning	Number of events
Lee et al., 2018 [27]	Logistic regression (nearest value) Logistic regression (nearest value, recording frequency) Logistic regression (nearest value; recording frequency; minimum, maximum, and mean with highest effect size)	Classic model Classic model Classic model	AUC-ROC Sensitivity Specificity Positive predictive value Negative predictive value
Wang et al., 2023 [28]	Logistic regression Gradient-boosted tree Deep learning	Classic model Machine learning Deep learning	AUC-ROC Sensitivity Precision Specificity Brier score Calibration plot
King et al., 2023 [29]	Regression model	Classic model	Positive predictive value Sensitivity Negative predictive value Specificity
Chang et al., 2019 [30]	Random forest XGBoost Artificial neural net Recurrent neural network—Long short-term memory network	Machine learning Machine learning Deep learning Deep learning	Heart rate AUC-ROC Blood oxygen level AUC-ROC Mean arterial pressure AUC-ROC Respiratory rate AUC-ROC Systolic blood pressure AUC-ROC
Bhattacharyya et al., 2022 [31]	Logistic regression Random forest Bidirectional long short-term memory	Classic model Machine learning Deep learning	AUC-ROC AUPRC Sensitivity Brier score
Xu et al., 2022 [32]	Logistic regression Decision tree Random forest	Classic model Machine learning Machine learning	AUC-ROC Sensitivity Specificity Accuracy Precision Positive predictive value Negative predictive value F-1 score
Ladios-Martin et al., 2020 [33]	Averaged perception Bayes point machine Decision tree Boosted decision forest Decision jungle Locally deep support vector machine Logistic regression Neural network Support vector machine	Machine learning Machine learning Machine learning Machine learning Machine learning Machine learning Classic model Deep learning Machine learning	AUC-ROC Sensitivity Specificity Accuracy Positive predictive value Negative predictive value
Toledo et al., 2024 [34]	Perceptron neural network 1 Perceptron neural network 2 Neural network with radial basis function Decision tree Random forest	Deep learning Deep learning Deep learning Machine learning Machine learning	R square Root mean square error Mean absolute error
Cui & Jin, 2023 [35]	Bidirectional long short-term memory Recurrent neural network Gated recurrent units Long short-term memory Random forest Logistic regression	Deep learning Deep learning Machine learning Machine learning Machine learning Classic model	F-1 score Precision Sensitivity AUC-ROC
Brandao-de-Resende et al., 2023 [36]	Logistic regression Decision tree Random forest XGBoost	Classic model Machine learning Machine learning Machine learning	Sensitivity Specificity
Cho et al., 2022 [37]	Neural network	Deep learning	Time for triage Intraclass correlation coefficient

**Table 4 nursrep-15-00055-t004:** Types of data.

Author(s)	Type of Data Used by AI	Input Data	Output Data
Abraham et al., 2023 [14]	Vitals Clinical notes	Structured data (continuous data) Unstructured data	Continuous data
Meyfroidt et al., 2011 [15]	Vitals Admission data (patient history and preoperative medical condition) Medication data Laboratory data Physiological data (urine output, ventilator, blood loss)	Structured data (continuous data) Unstructured data	Continuous data
Mutegeki et al., 2023 [16]	Vitals Demographical data Triage variables Disposition Chief patient complaint Past medical history Laboratory data Medication data Imaging data Historical usage statistics	Structured data (continuous, binary, categorical) Unstructured data	Categorical data
Nazzal et al., 2020 [17]	Vitals Demographical data Laboratory data Past medical history Medication data Admission data Discharge data	Structured data (continuous, binary, categorical) Unstructured data	Categorical data
De Koning et al., 2023 [18]	Vitals Demographical data Triage variables Past medical history Medication data ECG’s description Physical examination Symptoms	Structured data (continuous data) Unstructured data	Binary data
Zaboli et al., 2024 [19]	Vitals Demographical data Triage code Past medical history	Structured data (continuous, categorical) Unstructured data	Categorical data
Gu et al., 2024 [20]	Nursing notes	Unstructured data	Binary data, text
Horgn et al., 2017 [21]	Vitals Demographical data Chief complaint Nursing notes Acuity level	Structured data (continuous, binary, categorical) Unstructured data	Binary data
Ivanov et al., 2021 [22]	Vitals Demographical data Past medical history Chief complaint Nursing notes	Structured data (continuous, categorical) Unstructured data	Categorical data
Cramer et al., 2019 [23]	Vitals Demographical data Admission data Laboratory data Comorbidities Ventilation status	Structured data (continuous, categorical data)	Categorical data
Greenbaum et al., 2019 [24]	Vitals Demographical data Nursing notes	Structured data (continuous, categorical data)	Categorical data
Yu et al., 2020 [25]	Vitals Demographical data Admission data	Structured data (continuous, binary, categorical data)	Categorical data
Tsumoto et al., 2006 [26]	Type of near miss Patient factors Medical staff factors Shift information	Structured data (continuous, categorical data) Unstructured data	Continuous data
Lee et al., 2018 [27]	Vitals Demographical data Ventilation status Nurses’ notes	Structured data (continuous, categorical data) Unstructured data	Categorical data
Wang et al., 2023 [28]	Vitals Demographical data Medication data Laboratory data Comorbidities Glasgow coma scale	Structured data (continuous, binary, categorical data)	Continuous data
King et al., 2023 [29]	Rounding teams’ recordings	Unstructured data (audio)	Text
Chang et al., 2019 [30]	Vitals Demographical data Laboratory data Medication data Mortality	Structured data (continuous, categorical, binary data)	Continuous data
Bhattacharyya et al., 2022 [31]	Vitals Demographical data Laboratory data Medication data Ventilation status Sequential Organ Failure Assessment score	Structured data (continuous, categorical data) Unstructured data	Continuous data
Xu et al., 2022 [32]	Age Comorbidities Surgical history Ventilation status Glasgow coma scale	Structured data (continuous, binary, categorical data)	Categorical data
Ladios-Martin et al., 2020 [33]	Demographical data Mobility Care process Medication data Laboratory data Mental status Surgical history Skin status Nutrition	Structured data (continuous, categorical data) Unstructured data	Categorical data
Toledo et al., 2024 [34]	Demographical data Bed bath time Comorbidities Medication data Devices Oxygen therapy	Structured data (continuous, binary data)	Continuous data
Cui & Jin, 2023 [35]	Vitals Laboratory data Braden scale-related features Glasgow Coma Scale Overall patient condition	Structured data (continuous, categorical data) Unstructured data	Categorical data
Brandao-de-Resende et al., 2023 [36]	Demographical data History of presenting illness Medical history Signs and symptoms	Structured data (continuous, binary data) Unstructured data	Categorical data
Cho et al., 2022 [37]	Vitals Chief concern Medical history Allergies	Unstructured data (voice)	Text

## Data Availability

No new data were created or analyzed in this study. Data sharing is not applicable to this article.

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
