# Peer review of "Exploring Applications of Artificial Intelligence in Critical Care Nursing: A Systematic Review"

_nursrep, 2025, doi:10.3390/nursrep15020055_

Round 1

Reviewer 1 Report

Comments and Suggestions for Authors

Comments on the Quality of English Language

Well written 

Reviewer 2 Report

Comments and Suggestions for Authors

This manuscript gives a comprehensive systematic review on the applications of Artificial Intelligence (AI) in critical care nursing, focusing on health outcomes, AI model techniques, and data types used in critical care settings, exploring the effectiveness of various AI techniques in critical care units, emergency departments, operating rooms, and prehospital care, emphasizing how AI can support clinical decision-making and improve nursing-sensitive outcomes.

Major Strengths:

The manuscript addresses a relevant and current topic, as AI is increasingly being integrated into healthcare practices, including critical care settings.

The study was registered with PROSPERO and followed PRISMA guidelines, ensuring transparency and adherence to established systematic review standards.

The manuscript provides detailed tables summarizing AI models, data types, and performance metrics, contributing to the clarity and reproducibility of the review.

Major Concerns:

The background could be more focused on the role of AI specifically in nursing practice rather than general applications in healthcare.

Clarify the connection between AI and critical care nursing from the outset.

The manuscript interchanges terms like "classical models," "machine learning," and "deep learning" without consistent definitions.

Define each category clearly and maintain consistency throughout.

The inclusion and exclusion criteria should be clearly presented in a separate section rather than being embedded in other paragraphs.

Create a separate subsection under "Methods" for clarity.

Tables 1-4 are informative but lack clarity in differentiating structured vs. unstructured data usage.

Include a summary table highlighting the proportion of studies using structured vs. unstructured data.

The discussion section lacks a critical appraisal of the limitations of AI models, such as potential biases in the datasets and challenges in clinical implementation.

Add a dedicated paragraph discussing the limitations and ethical considerations of AI use in critical care.

The conclusion effectively summarizes the findings but could be strengthened by offering specific recommendations for clinical practice and future research.

Minor grammatical inconsistencies were noted, such as "the AI techniques varies" (should be "vary").

Comments on the Quality of English Language

The manuscript demonstrates a good command of English but would benefit from minor grammatical refinements and clarification of technical terminology to  more clearly express the research.

Author Response

Please, see the attched file.

Round 2

Reviewer 1 Report

Comments and Suggestions for Authors

Thanks for addressing the comments

Author Response

Thanks again to the reviewer. 

Reviewer 2 Report

Comments and Suggestions for Authors

I have revised this manuscript for the second time, substantial and good changes have been introduced to the manuscript, but I have some concerns before it is considered for publication, which I list below.

Introduction

Strengthen the theoretical background with a broader discussion of gaps in the current literature. Include additional citations highlighting the advantages and limitations of the most common AI techniques used in the critical care setting.

Methods

Provide a more detailed description of the search strategy and exclusion criteria. Indicate the number of reviewers involved and the method of conflict resolution.

Results

Integrate details in the tables, specifying the analysis methods applied for each included study. Improve the graphical layout of the PRISMA diagram for optimal readability.

Discussion and Conclusion

Compare AI techniques in more depth, highlighting specific advantages for nursing practice. Discuss more fully the methodological limitations of the included studies, such as the lack of unstructured data.
